AI in dermatology: a comprehensive review into skin cancer detection

http://orcid.org/0000-0002-8877-5630 Behara Kavita 1 beharak@mut.ac.za
http://orcid.org/0000-0002-6349-5907 Bhero Ernest 2
Agee John Terhile 2
1 Department of Electrical Engineering, Mangosuthu University of Technology , Durban, Kwazulu- Natal , South Africa
2 Discipline of Computer Engineering, University of KwaZulu Natal , Durban, KwaZulu-Natal , South Africa
Moparthi Nageswara Rao
Electronic publication date: 2024 Dec 5
Publication date: 2024
Volume: 10
Electronic Location ID: e2530
Received 2024 Jul 4; Accepted 2024 Oct 28
Copyright: © 2024 Behara et al.
Copyright year: 2024
Copyright holder: Behara et al.
License: This is an open access article distributed under the terms of the Creative Commons Attribution License, which permits unrestricted use, distribution, reproduction and adaptation in any medium and for any purpose provided that it is properly attributed. For attribution, the original author(s), title, publication source (PeerJ Computer Science) and either DOI or URL of the article must be cited.
License URL: https://creativecommons.org/licenses/by/4.0/

Keywords: Artificial intelligence, Classification, Deep learning, Dermatology, Image preprocessing, Machine learning, Skin cancer, Clinical decisions, Prediction

Funding: The authors received no funding for this work.

==============================
Background

Artificial Intelligence (AI) is significantly transforming dermatology, particularly in early skin cancer detection and diagnosis. This technological advancement addresses a crucial public health issue by enhancing diagnostic accuracy, efficiency, and accessibility. AI integration in medical imaging and diagnostic procedures offers promising solutions to the limitations of traditional methods, which often rely on subjective clinical evaluations and histopathological analyses. This study systematically reviews current AI applications in skin cancer classification, providing a comprehensive overview of their advantages, challenges, methodologies, and functionalities.

Methodology

In this study, we conducted a comprehensive analysis of artificial intelligence (AI) applications in the classification of skin cancer. We evaluated publications from three prominent journal databases: Scopus, IEEE, and MDPI. We conducted a thorough selection process using the PRISMA guidelines, collecting 1,156 scientific articles. Our methodology included evaluating the titles and abstracts and thoroughly examining the full text to determine their relevance and quality. Consequently, we included a total of 95 publications in the final study. We analyzed and categorized the articles based on four key dimensions: advantages, difficulties, methodologies, and functionalities.

Results

AI-based models exhibit remarkable performance in skin cancer detection by leveraging advanced deep learning algorithms, image processing techniques, and feature extraction methods. The advantages of AI integration include significantly improved diagnostic accuracy, faster turnaround times, and increased accessibility to dermatological expertise, particularly benefiting underserved areas. However, several challenges remain, such as concerns over data privacy, complexities in integrating AI systems into existing workflows, and the need for large, high-quality datasets. AI-based methods for skin cancer detection, including CNNs, SVMs, and ensemble learning techniques, aim to improve lesion classification accuracy and increase early detection. AI systems enhance healthcare by enabling remote consultations, continuous patient monitoring, and supporting clinical decision-making, leading to more efficient care and better patient outcomes.

Conclusions

This comprehensive review highlights the transformative potential of AI in dermatology, particularly in skin cancer detection and diagnosis. While AI technologies have significantly improved diagnostic accuracy, efficiency, and accessibility, several challenges remain. Future research should focus on ensuring data privacy, developing robust AI systems that can generalize across diverse populations, and creating large, high-quality datasets. Integrating AI tools into clinical workflows is critical to maximizing their utility and effectiveness. Continuous innovation and interdisciplinary collaboration will be essential for fully realizing the benefits of AI in skin cancer detection and diagnosis.

Introduction

Skin cancer is the most frequently diagnosed cancer globally, with millions of new cases reported each year. Non-melanoma skin cancers, including basal cell carcinoma (BCC) and squamous cell carcinoma (SCC), are the most prevalent. While melanoma accounts for a smaller proportion of cases, it is responsible for the majority of skin cancer-related deaths (Asiri et al., 2023). The rising incidence of skin cancer worldwide is attributed to increased sun exposure, ageing populations, and advances in detection methods (Ameri, 2020). Several factors elevate the risk of developing skin cancer, the most significant being excessive exposure to ultraviolet (UV) radiation from the sun or tanning beds. Additional risks include having fair skin that burns easily, a history of sunburn, and a family history of skin cancer (Behara, Bhero & Agee, 2023). Other contributing factors include older age, male gender, the presence of multiple or atypical moles, immunosuppression, and geographic location, with regions of intense sun exposure—such as Australia and the southern United States—reporting higher incidence rates. Occupational exposure to specific chemicals and radiation further increases the risk of skin cancer (Diab, Fayez & El-Seddek, 2022). Diagnosing skin cancer involves clinical expertise combined with histopathological examination (Madarkar & Koti, 2021).

Clinicians rely on a combination of visual inspection, dermoscopic analysis, and patient history to distinguish between malignant and benign lesions. However, this process can be subjective and vary significantly among practitioners (Jojoa Acosta et al., 2021). Although histopathological examination, the gold standard for diagnosis, is highly accurate, it requires a biopsy and microscopic analysis of skin tissue, making it invasive and time-consuming and dependent on the availability of specialized pathology services (Behara, Bhero & Agee, 2024a). Traditional diagnostic methods like visual inspection and biopsy come with inherent limitations. Visual inspection, which depends heavily on the clinician’s experience, can lead to misdiagnosis or delays, especially when assessing atypical lesions or those located in less accessible areas (Jain et al., 2021). While biopsy provides a definitive diagnosis, it is invasive and may cause discomfort and scarring for the patient.

Additionally, the turnaround time for biopsy results can delay the initiation of treatment (Behara, Bhero & Agee, 2024b). Traditional methods also face accessibility challenges, particularly in resource-limited settings where specialized dermatological and pathological services are scarce. It underlines the need for more efficient, accurate, and non-invasive diagnostic tools, such as AI technologies (Javaid et al., 2022).

Early detection of skin cancer is crucial for effective treatment and significantly improves survival rates. For instance, melanoma, a particularly aggressive form of skin cancer, has a five-year survival rate of about 99% when detected early. Still, this rate declines once it spreads to distant organs (American Cancer Society, 2022).

Artificial Intelligence (AI) is a multidisciplinary field that facilitates the development of cost-effective, widely accessible, and highly efficient tools for various tasks, significantly reducing processing time and costs while boosting accuracy (Kaplan & Haenlein, 2020). Advancements from the fourth industrial revolution have enabled AI systems to perform tasks that traditionally require human intelligence, utilizing Big Data for self-learning and sophisticated behaviors (Farhud & Zokaei, 2021).

Dermatology has greatly benefited from AI, transforming many aspects of diagnostic processes into digital, automated systems (Sturmberg & Bircher, 2019; Carayon et al., 2018). AI reduces manual administrative tasks, allowing doctors to focus on critical medical decisions (Xie et al., 2020). AI-based diagnostic systems are particularly effective for early skin cancer detection, using medical data and analytics to improve processes and simplify medical services (Dhieb et al., 2019).

The use of electronic health records (EHRs), medical imaging, and monitoring devices has significantly expanded the volume of medical data collected. AI leverages this data to generate valuable insights for medical interventions (Pee, Pan & Cui, 2018). Machine learning techniques, supported by advanced data storage and processing capabilities, are central to achieving these outcomes (Istepanian & Al-Anzi, 2018; Gandomi, Chen & Abualigah, 2022).

The COVID-19 pandemic has highlighted the importance of integrating AI and the Internet of Things (IoT) in dermatology (Tobore et al., 2019; Khan et al., 2022). AI has broad applications in oncology, particularly in the detection and treatment of cancer. Accurate diagnosis of skin cancer, especially in cases exacerbated by high ultraviolet radiation levels, is critical due to the visual similarity of various lesions, requiring the expertise of skilled dermatologists (Zakhem et al., 2021; Gershenwald et al., 2017).

Dermoscopy, while more accurate than unaided visual inspection, has an accuracy rate of around 80% under typical clinical conditions (Dinnes et al., 2018). AI can complement dermatologists by enhancing diagnostic accuracy, particularly in early melanoma detection, which improves patient survival rates (Dinnes et al., 2018). AI’s capabilities include providing diagnostic support, facilitating medical interventions, and helping develop proactive health strategies (Shahin et al., 2023).

Many modern hospitals are incorporating AI technologies to reduce operational costs and improve diagnostic precision (Gershenwald et al., 2017). AI systems can analyze large volumes of medical data rapidly, identifying patterns and anomalies that may elude human detection (Thurnhofer-Hemsi et al., 2021). This streamlines workflows, allowing healthcare professionals to focus more on patient care while ensuring more accurate diagnoses. Additionally, AI can provide detailed treatment options, empowering clinicians to make more informed decisions (Dinnes et al., 2018).

AI systems also offer personalized treatment recommendations based on a patient’s medical history and current condition, leading to improved health outcomes (Alowais et al., 2023; Esmaeilzadeh, Mirzaei & Dharanikota, 2021). These technologies can monitor patient progress and predict potential complications, enabling early intervention and better long-term health management (Khalid et al., 2023).

Article motivation

AI holds great promise for the future of healthcare, but its integration into medical diagnostics, mainly through machine learning (ML) and deep learning (DL), presents several challenges. These include potential system failures that could harm patient records, privacy concerns that limit data access, and ethical, legal, and medical issues surrounding AI’s role in health decisions (Dinnes et al., 2018). Despite these obstacles, AI supports preventative care by utilizing Big Data to conduct extensive comparative analyses, helping medical professionals identify patterns that enhance diagnostics and intervention efforts (Yousef, Kassem & Hosny, 2023).

In recent years, AI has revolutionized skin cancer diagnosis. Smartphone apps can analyze photos of suspicious moles and quickly suggest whether medical attention is needed. Handheld AI-powered devices assist dermatologists by enlarging lesions and providing immediate feedback, reducing the likelihood of misdiagnosis (Dinnes et al., 2018; Yousef, Kassem & Hosny, 2023). Wearable AI devices further contribute by monitoring skin lesions and alerting patients and doctors to any changes (Wan et al., 2023). Importantly, AI complements, rather than replaces, dermatologists, enhancing skin cancer diagnosis and treatment. In this AI-driven era, early detection offers hope for improved skin cancer prevention (Zhou et al., 2017).

AI-driven diagnostic tools leverage machine learning algorithms, including deep learning, to analyze and classify data from sources such as electronic health records, images, and videos (Asiri et al., 2023). Machine learning methods include regression, classification, and clustering, which can handle data types ranging from text and numbers to images and videos (Mbunge & Batani, 2023). Classification algorithms, such as neural networks, decision trees, and Bayesian networks, can be trained using vast datasets (Ichim & Popescu, 2020). Supervised learning uses labeled data for training, while unsupervised learning works without class labels, creating prediction models based on historical data.

While recent reviews have highlighted advancements in AI for medical applications, including skin cancer detection (Deepa, ALMahadin & Sivasamy, 2024; Gohil & Desai, 2024), there remains a gap in addressing the specific challenges and methodologies required for full integration of AI systems into clinical workflows. Many studies emphasize AI’s diagnostic accuracy but fail to explore its practical implementation across diverse healthcare settings comprehensively. Furthermore, the existing literature lacks an in-depth evaluation of the functionalities AI systems provide beyond diagnosis, such as their contributions to improving healthcare delivery and patient outcomes (Strzelecki et al., 2024; Furriel et al., 2024).

The advancements in AI for medical diagnostics, particularly skin cancer detection (Deepa, ALMahadin & Sivasamy, 2024; Gohil & Desai, 2024), has garnered considerable attention due to its potential to improve detection accuracy and efficiency. Several studies have examined various AI models, providing valuable insights into machine learning and deep learning techniques for skin lesion classification (Wu et al., 2020). However, much of the existing literature has focused narrowly on the technical performance of AI models, often overlooking critical challenges related to data scarcity, the underrepresentation of diverse skin tones, and the practical implementation of these technologies in low-resource settings (Strzelecki et al., 2024; Furriel et al., 2024).

This review offers a novel contribution by not only examining the technical performance of AI-based skin cancer detection but also addressing critical underexplored areas such as data disparities across different skin types, ethical concerns regarding AI deployment, and the practical challenges of implementing AI in dermatological care within regions with limited healthcare infrastructure. Furthermore, this review goes beyond diagnostic accuracy by exploring the broader functionalities of AI systems, such as their potential to streamline healthcare delivery, enhance patient outcomes, and provide equitable care across diverse populations.

This review aims to systematically analyze the applications of AI in skin cancer detection, focusing on recent advancements and their practical implications for clinical practice. The key objectives of the review are:

RQ1: What are the key advantages of AI integration in skin cancer detection?

RQ2: What challenges and difficulties exist in implementing AI in dermatology?

RQ3: What methodologies are commonly employed in AI-based skin cancer detection?

RQ4: What functionalities do AI systems offer beyond diagnostic accuracy, and how do they contribute to healthcare delivery?

The rest of the article is structured as follows: ‘Survey Methodology’ provides the methodology used for the comprehensive review and discusses the classification framework using four dimensions, namely advantages, difficulties, methodology, and functionality classification framework. Finally, ‘Conclusions’ is discussed.

Survey methodology

The novelty of this review article lies in its structured classification framework, which goes beyond AI’s diagnostic accuracy to focus on its integration into clinical workflows. By categorizing AI’s role into advantages, challenges, methodologies, and functionalities, the review provides a comprehensive perspective on bridging gaps in dermatology. Using PRISMA guidelines, it systematically analyzes literature from 2019–2023, ensuring a transparent review process. The article identifies critical gaps, particularly in AI integration challenges, and calls for future research in areas like data privacy and generalization across diverse populations. It offers a roadmap for enhancing AI’s role in dermatology with a systematic, forward-looking approach.

The literature review consists of three essential phases: (1) planning and conducting the review, (2) reporting the results of the literature review, and (3) scoping the insights from the AI literature in skin cancer diagnosis using the classification framework of advantages, difficulties, methodology, and functionality for potential implementation and development of AI tools and methodologies. The Preferred Reporting Items for Systematic Reviews and Meta-Analysis (PRISMA) guidelines (Zhou et al., 2017) were used to develop literature search strategies and quantitative research methods. Figure 1 illustrates the selection study process using a PRISMA flowchart.

Figure 1 PRISMA flow diagram for systematic review process.

Planning and conducting the literature review

The first phase of our literature review includes planning the literature review, which consists of five crucial steps, as shown in Fig. 1. The five steps include (1) Search Strategy, (2) Inclusion Criteria, (3) Exclusion Criteria, and (4) Data Extraction, Synthesis, and Analysis. The research questions RQ1, RQ2, RQ3, and RQ4 guide the planning phase.

Search strategy

The initial search approach involved searching for relevant journal articles in Scopus, IEEE, and MDPI using the Mendeley Repository. These databases were selected for their focus on high-impact, peer-reviewed research in technology, engineering, and medical fields, particularly those related to AI and its applications in skin cancer detection. The search thread was “Artificial Intelligence AND Skin Cancer AND Deep Learning OR Machine Learning AND Image Processing”. It is essential to acknowledge that within a search thread, the Boolean logical operators AND & OR are utilized to establish connectedness. The keywords and research questions were constructed according to the PICOS (population, intervention, comparison, outcomes, and study design) framework as follows:

P – Studies that focus solely on the application of AI to skin cancer.

I – Studies that examined theoretical, statistical, or mathematical frameworks for skin cancer-related AI applications or systems.

C – Comparison with alternative treatments/AI models.

O – Histopathology of skin cancer lesions. Studies on using AI for detecting, classifying, or diagnosing skin cancer.

S – Investigate studies proposing a novel AI-based approach to detecting skin cancer.

There were no guidelines regarding the permitted publishing location, study type, or control group. The backward snowball technique was also used to find articles that were not found using the automated research strategy.

Inclusion criteria

The field of artificial intelligence is advancing rapidly. To ensure our literature review remains current and relevant, we established inclusion criteria that consider the following: journal articles published from 2019 to 2023, academic research conducted in English, and only original articles selected for analysis. These articles represent primary research contributions rather than secondary summaries or reviews. By focusing on journal articles published within this time frame, the literature review encompasses recent research and developments in AI. Limiting the scope to English-language articles ensures consistency and facilitates cross-referencing across studies. Adhering to these inclusion criteria allows the review to capture relevant and up-to-date insights from the AI research landscape.

Exclusion criteria

Studies excluded from this review include publications such as academic reports, case studies, surveys, review articles, and dissertations. Articles that do not focus on skin cancer or use non-AI-based approaches for skin cancer diagnostics were also excluded. These criteria were established to maintain the quality and relevance of the studies included in the systematic review.

Data extraction, synthesis, and analysis

The search results were reviewed by three experts who independently screened the titles and abstracts. After screening selected studies for pertinent data, the authors evaluated the full texts of the original articles. Disagreements were settled by back-and-forth dialogue and mutual discussion. The relevant information on authors, publication year, study design, participants, interventions, and outcomes was extracted from the selected articles. The findings were subsequently summarised based on a classification framework that includes advantages, difficulties, methodology, and functionality. The extracted studies were synthesized using a narrative method that summarizes and integrates findings from the included studies, providing a comprehensive overview of the current state of AI models in skincare.

The process of systematic review using the classification framework

By following these detailed steps, the literature review ensures a rigorous and systematic approach to identifying, evaluating, and synthesizing relevant research findings in AI-driven skincare. Specifically, we first organized the research into four distinct dimensions in the established classification system. Relevant factors were found for each dimension, as shown in Fig. 2.

Figure 2 The process of systematic review using the classification framework.

1. Advantages: The application of AI offers significant benefits, including automated decision-making, enhanced patient monitoring (particularly for older patients), early diagnosis, and process optimization.

2. Difficulties: Organizations may face several challenges when implementing AI, such as data-related issues, including digitization, consolidation, and availability. Additionally, there are privacy and legal challenges, along with government regulations. Patient-related challenges involve human interventions, treatment, judgment, and data errors.

3. Methodologies: This dimension includes adopting AI-based models, data processing techniques, machine learning algorithms, and expert systems.

4. Functionalities: This dimension refers to the features that organizations can access through AI. For clinics, AI enhances clinical decision-making, provides up-to-date information, optimizes resource allocation, and facilitates information sharing. AI supports diagnosis, treatment, consultations, and continuous monitoring of patients. Additionally, at the sector level, AI functionalities include IoT integration, data collection, medical imaging, research development, and remote surgery capabilities.

Results of the literature review

The search process involved both automated and manual methods. Initially, five databases were explored, yielding 1,156 unique articles. After applying a Boolean logical filtering technique, this number was reduced to 691. A detailed screening process was then conducted to remove duplicates, resulting in the elimination of 102 articles. Researchers then manually reviewed the remaining articles, focusing on those most relevant to the study’s conceptual and empirical aspects. This manual screening of titles and abstracts further reduced the number of articles to 206. The researchers then conducted a full review of these articles, evaluating criteria such as the research aims, questions, data description, methodology, data analysis approach, and presentation of findings. Following this thorough review, 137 articles were excluded due to irrelevance, leaving 65 articles. An additional 42 articles were included through the backward snowball method, bringing the total to 111. Finally, 16 articles were excluded based on a quality assessment, leaving 95 articles for analysis. The journal’s search methodologies for the review are presented in Table 1 and Fig. 3.

Table 1 Journal search methodology.

	Search methodology	Final search results	
Auto-search strategy	Manual search strategy	Backward snowball strategy	Total	
Keywords	Filters	Title	Abstract	Reading full article	
Scopus	543	268	115	91	28	49	37	
IEEE	333	227	119	69	21	47	34	
MDPI	280	196	84	46	16	15	24	
Total	1,156	691	318	206	65	111	95	

Figure 3 Journals review search status.

Article distribution by year of publication

AI has significantly advanced the detection and treatment of skin cancer in recent years. Notably, many recent studies reflect AI’s growing global acceptance and implementation in chronic disease identification. We reviewed articles published within the last five years to ensure our study remains current. Figure 4 illustrates the number of articles published from 2019 to 2023 based on the keyword search discussed in ‘Inclusion Criteria’.

Figure 4 Distribution analysis of reviewed journals per annum.

Database distribution of articles

The Pareto analysis distribution of the chosen articles by database source is shown in Fig. 5. Thirty-seven publications were found in the SCOPUS database, 34 in the IEEE, and 24 in the MDPI database. The top publications were from SCOPUS and IEEE.

Figure 5 Pareto analysis of journal publication.

Classification framework

This section presents a review of the classification framework, encompassing the following dimensions: advantages, difficulties, approaches, and functionality. This framework scopes insights from the AI literature on skin cancer detection, providing a structured approach to organize and understand current research. Appendix 1 presents the classification framework and its various dimensions, which assisted our systematic literature review. Detailed studies of each dimension of the classification framework are discussed in subsequent subsections.

Dimension 1: Advantages of AI integration in skin cancer detection

This dimension explores the advantages of AI in skin cancer diagnostics, focusing on four key aspects: (1) early detection, (2) enhanced diagnostic accuracy, (3) accessibility to expert clinicians, and (4) efficiency.

Early detection

AI systems have significantly improved the analysis and detection of skin lesions, distinguishing minute changes in texture, color, and shape that might be missed by the human eye (Yu et al., 2019; Moldovanu et al., 2021). Early detection, especially of melanoma, greatly enhances survival rates, with studies showing a 99% survival rate when caught in the early stages (Shinde et al., 2023; Mridha et al., 2023). AI technologies, including computer vision and machine learning algorithms, facilitate the early identification of abnormalities, enabling timely intervention (Priyadharshini et al., 2023).

In dermatology, AI plays a crucial role in extracting data from digitized images, enhancing early diagnosis and treatment. Biasi et al. (2022) emphasize the need for continuous training-test iterations and flexible system architectures to address transfer learning challenges. AI-powered tools also provide real-time screening and diagnostic support during patient examinations, improving the accuracy and efficiency of clinical workflows (Melarkode et al., 2023; Codella et al., 2018). Total Body Imaging Systems, integrated with AI, further enhance early detection for high-risk patients by monitoring changes across larger surface areas (Korotkov et al., 2019; Strzelecki et al., 2021).

Improved diagnostic accuracy

AI algorithms provide reliable, impartial judgments, reducing the risk of human error due to fatigue or bias. These systems enhance diagnostic accuracy by integrating clinical data, such as patient history and demographics, with image analysis (Maqsood & Damaševičius, 2023). Machine learning (ML) tools are critical in clinical decision support, offering personalized treatment options and improving healthcare efficiency (Javaid et al., 2022). Han et al. (2022) found that AI-assisted diagnostic accuracy significantly improved non-expert physicians’ performance in real-world settings, demonstrating AI’s potential to enhance clinical decision-making (Strachna & Asan, 2020).

Advanced AI methods, such as support vector machines (SVM) and deep learning models like ResNet-50, offer robust tools for melanoma detection, improving diagnostic accuracy by combining conventional image processing with deep learning (Hagerty et al., 2019). Studies have shown that AI can achieve diagnostic performance comparable to or better than experienced dermatologists, using metrics such as sensitivity, specificity, and AUC-ROC (Esteva et al., 2017).

Access to expert clinicians

AI-driven systems expand access to dermatological expertise, particularly in underserved or remote areas. Patients can upload images of their skin issues for remote review via AI-powered telemedicine platforms, facilitating early intervention and improved outcomes (Carvalho et al., 2021; Hossain, Ferdousi & Alhamid, 2020). AI systems can also be deployed in rural areas, providing high-quality diagnostic services where access to specialized medical professionals is limited. Teledermatology platforms with integrated AI analyze patient-submitted images and provide diagnostic insights, reducing the need for in-person visits.

Efficiency gains in diagnosis

AI systems rapidly analyze large volumes of skin images, optimizing diagnostic processes and reducing patient waiting times. This increased efficiency enhances clinical workflows and healthcare delivery (Esteva et al., 2017; Yehia et al., 2019). Lucieri et al. (2022) developed ExAID, an explainable AI framework for skin lesion diagnosis, which uses concept activation vectors and concept localisation maps to offer comprehensible explanations. ExAID achieves up to 81.46% accuracy on dermoscopic datasets, aiming to improve transparency and trust in AI-based systems for clinical use. Gu et al. (2020) address domain shift issues in skin disease classification, using progressive transfer learning and adversarial domain adaptation to improve recognition performance across clinical settings, enhancing the robustness of AI systems for melanoma detection.

Dimension 2: Challenges and difficulties in AI integration for skin cancer detection

Using AI in healthcare presents several challenges that could hinder its adoption. Key issues include data availability, consolidation, and digitization (Kaplan & Haenlein, 2020; Carayon et al., 2018). Privacy and legal concerns, including regulations and legislation, further complicate implementation (Lohachab, Lohachab & Jangra, 2020; Pee, Pan & Cui, 2018). Patient-related difficulties such as human interventions, data errors, and judgment errors also pose significant challenges (Pee, Pan & Cui, 2018; Istepanian & Al-Anzi, 2018).

Patient safety-related obstacles

Machine learning, natural language processing, and expert systems rely on medical data to create decision-support models. However, inaccuracies in hospital data can lead to harmful diagnosis errors (Xie et al., 2020). Data errors remain a significant issue when using AI to process skin lesion data (Gandomi, Chen & Abualigah, 2022; Tobore et al., 2019). Machine learning models may make incorrect decisions if the data is unreliable, highlighting the need for high-quality data (Hwang et al., 2016; Khater et al., 2023).

Variation in system effectiveness

AI-based skin lesion analysis systems can detect multiple types of skin cancer, but their effectiveness varies depending on the lesion type. Systems for detecting melanoma tend to be more accurate due to the availability of large, well-annotated datasets. At the same time, those for less common cancers like Merkel cell carcinoma or basal cell carcinoma (BCC) show reduced sensitivity (Korotkov et al., 2019). It emphazises the need for diverse datasets and advanced algorithms to improve clinical applicability across various cancer types. Deep learning models generally outperform classical systems in complex lesion detection tasks.

Data legal issues

AI systems require large amounts of sensitive medical data for training and validation. Ensuring data confidentiality is essential for maintaining patient trust and complying with legal standards. Robust encryption and anonymization techniques are necessary to protect patient information (Rieke et al., 2020). In addition, healthcare AI systems must adhere to regulatory and ethical guidelines set by bodies like the FDA and EMA to ensure patient safety, data integrity, and transparency in AI decision-making (European Commission, 2020).

Collaboration between AI and clinicians

Despite AI advancements, clinicians remain the ultimate authority in medical decision-making. Effective collaboration between doctors and AI systems is crucial, as relying solely on AI can lead to erroneous diagnoses and treatment outcomes (Maqsood & Damaševičius, 2023; Kumar et al., 2023; Munappy et al., 2022).

Data privacy concerns

The usage, distribution, and retrieval of patient data raise legitimate privacy concerns, especially in the context of AI and cloud computing used in healthcare (Kumar et al., 2023). While these technologies offer significant benefits, they also present challenges related to security, privacy, cybersecurity, and ethics. Many hospitals and governmental organizations have protocols in place for ethical data collection and exchange.

Healthcare data gathering requires government approval (Abdar et al., 2021; Meena & Hasija, 2022). Ethical concerns surrounding AI include issues related to inequality, unemployment, biases, and regulatory approaches (Mahbod et al., 2020). Studies have proposed solutions such as rewarding ethical hacking and ensuring robust systems to minimize the ethical risks associated with AI in healthcare (Patel et al., 2021; Jiang, Li & Jin, 2021).

AI models depend on large, diverse datasets to generalize effectively. In skin cancer detection, this involves access to dermoscopic images that represent a variety of skin types, lesions, and demographic backgrounds. However, privacy concerns, data-sharing restrictions, and labeling challenges make obtaining comprehensive datasets difficult (Murphy et al., 2021). While AI has improved diagnostic accuracy and efficiency, issues with data privacy and availability continue to hinder its broader adoption in clinical practice.

Governments have raised concerns over automated decision-making in healthcare and its impact on patient rights (Jain et al., 2021). These concerns lead to restrictions on data collection, processing, and use. Researchers must prioritize data quality, testing, and proper documentation before deploying AI systems (Hossain, Ferdousi & Alhamid, 2020; González-Cruz et al., 2020).

Generalization across diverse populations

A significant challenge AI models face is the lack of diverse representation in datasets due to privacy constraints, which prevents access to large-scale international databases. Studies have shown that the generalization capability of these models across various demographics—especially with respect to age, skin tone, and lesion type—remains limited. This bias occurs because data from underrepresented groups is either unavailable or restricted, leading to models that perform well in specific populations but fail in others (Murphy et al., 2021). It is a critical issue in skin cancer detection, where early and accurate diagnosis across all skin tones is crucial (Behara, Bhero & Agee, 2024a).

Data fragmentation and bias

Data fragmentation across institutions further complicates AI development. Medical data is often siloed within healthcare systems, limiting the AI model’s access to comprehensive and diverse datasets. This fragmentation introduces bias into the model, as it is trained on isolated and possibly skewed data, which reduces its reliability when applied in real-world clinical settings (Strzelecki et al., 2024). As a result, the lack of data standardization across institutions and regions exacerbates the challenges of creating a broadly applicable AI system.

Data security and ethical challenges

While ensuring patient privacy is paramount, overly stringent anonymization processes may strip crucial contextual information, such as genetic predisposition or medical history, which are essential for creating reliable AI models. This loss of detail can undermine the model’s diagnostic accuracy, particularly in complex cases where patient-specific factors are critical (European Commission, 2020). Moreover, ethical concerns arise regarding the appropriate use of sensitive patient data, especially in the development of commercial AI products (Rieke et al., 2020).

Data integration difficulties

AI methodologies require substantial data for effective processing, but collecting patient data poses ethical challenges. While classification and clustering techniques can achieve high accuracy with limited datasets, their practicality may be questionable in broader contexts (Hekler et al., 2019). A common issue in AI research is the reliance on proprietary datasets, which vary significantly in size and can affect the reliability and generalizability of AI models. Smaller datasets containing fewer than 1,000 images may limit a model’s ability to detect rarer forms of skin cancer (Birkenfeld et al., 2020). Conversely, larger datasets like HAM10000, with over 10,000 images, produce more reliable and generalizable results. It is crucial to acknowledge these limitations, as variability in skin types, lighting, and lesion presentation can significantly impact diagnostic outcomes in clinical settings.

Preprocessing collected data is essential for AI, especially when dealing with text data that requires natural language processing. Integrating various data types, including text, images, and videos, remains a challenge in medical data analysis (Yehia et al., 2019; Jin et al., 2019). Additionally, obtaining precise and reliable data from diverse sources, such as medical imaging and numerical data, poses significant challenges (Mobiny, Singh & Van Nguyen, 2019; Bozsányi et al., 2021). Han et al. (2022) identified a significant challenge in AI systems: a drop in accuracy when all top AI predictions were incorrect. It presents the importance of integrating AI with human expertise to mitigate potential errors and highlights the need for continuous evaluation and improvement to ensure AI reliability in clinical settings. Nigar et al. (2022) also noted challenges with their XAI-based system, particularly regarding the validation of LIME-generated explanations and the need for further testing to ensure model generalizability across diverse populations. Integrating such systems into clinical workflows and ensuring usability for medical practitioners remain vital obstacles. Afifi, GholamHosseini & Sinha (2019) focused on the difficulties of implementing SVM classifiers on hardware platforms, emphasizing the need for optimized hardware/software co-design to handle complex computations efficiently. They stressed the importance of advanced methodologies to balance performance with resource consumption, particularly when implementing SVM on FPGA systems.

The integration of AI tools into clinical workflows requires careful design to complement, rather than disrupt, routine practices. It involves ensuring user-friendly interfaces, providing adequate training for healthcare providers, and aligning AI systems with clinical protocols. Additionally, AI solutions must be interoperable with existing healthcare infrastructure, such as electronic health records (EHR) systems and diagnostic tools. Achieving interoperability requires standardized data formats and communication protocols, which can be challenging given the diversity of systems used in healthcare (Mandl & Kohane, 2012).

Classification method’s difficulties

Classifying skin cancer remains challenging due to variations in image resolution and difficulties distinguishing between different cancer types. Over the years, various AI techniques, including ML and DL, have been developed to improve skin lesion classification (Sharma et al., 2022). However, these techniques still face significant challenges, particularly when it comes to compatibility with low-resource devices and the complexity of models like YoloV5 and ResNet, which require substantial computational power (Yousef, Kassem & Hosny, 2023; Elshahawy et al., 2023). The “black box” nature of these models further complicates their adoption, as it is difficult to interpret how they generate predictions (Wang et al., 2022).

Additionally, models like SkinNetX, which rely on pre-trained algorithms, are challenging to implement in resource-constrained settings and may struggle to generalize findings to rare or unfamiliar lesions (Ogundokun et al., 2023). While AI offers valuable support, clinicians retain ultimate authority in decision-making, and collaboration between AI and human expertise is essential to avoid erroneous diagnoses (Maqsood & Damaševičius, 2023).

Efforts to address dataset limitations include frameworks like TED-GAN, which generate realistic skin lesion images to enhance classification accuracy, especially when data is scarce (Ahmad et al., 2021). Similarly, Zhao et al. (2021) proposed a model combining GANs and DenseNet201 to mitigate data insufficiency and class imbalance. Nigar et al. (2022) introduced an explainable AI system validated on the ISIC 2019 dataset, which significantly improved classification accuracy and trust in clinical settings through visual explanations.

Bias in AI models remains a concern, especially when they are trained on non-representative datasets. Models that predominantly use fair-skinned images may perform poorly on darker-skinned individuals. Addressing these biases requires curating diverse datasets and implementing generalizable techniques (Buolamwini & Gebru, 2018).

Verification methods for AI algorithms

One of the key challenges in evaluating AI algorithms for skin lesion detection is ensuring the reliability and generalizability of the results across different populations and clinical settings. A variety of verification methods have been employed in the studies reviewed, including cross-validation, external validation with independent datasets, and the use of hold-out test sets. Cross-validation, while useful for internal testing, often fails to demonstrate how well a model will perform on new, unseen data. Therefore, studies that incorporate external validation, using datasets collected independently of the training data, are critical in assessing a model’s generalizability. Additionally, the methods used to verify these algorithms vary widely, with some studies relying on manual annotation by dermatologists and others employing automated evaluation techniques. This variability in validation methods makes it difficult to directly compare the performance of different models. It is imperative that future studies adopt standardized validation procedures, using diverse and independent test datasets that include a variety of skin types, lesion types, and image acquisition settings, to ensure the robustness and reliability of AI models for clinical application (Strzelecki et al., 2021). Metrics such as accuracy, sensitivity, and AUC are commonly employed, but these may not fully capture clinical efficacy, particularly in settings where false negatives could have severe consequences.

Dimension 3: Methodologies in AI-based skin cancer detection

Medical data in AI applications can include images, quantitative data, or text, with image, video, and virtual reality processing being the most common approaches for image data analysis (Rahman et al., 2020). AI can make independent inferences with minimal human intervention (Weng & Zhu, 2021; Zhou et al., 2023). Studies show that AI has the potential to support and enhance capabilities in various medical fields, including skin cancer detection. Skin cancer places a significant burden on diagnostic systems due to the continuous care required, leading to high financial and time costs for patient consultations (Yousef, Kassem & Hosny, 2023). Many of these visits are unnecessary, squandering valuable resources.

The advancements of AI in dermatology offer notable improvements in diagnostic accuracy and operational efficiency compared to traditional methods. These improvements are particularly evident when considering factors such as speed, scalability, and bias as depicte in Table 2. While traditional methods still play a crucial role in clinical settings, AI has shown potential to transform skin cancer detection practices.

Table 2 Analysis of AI models based on ten criteria for skin cancer detection.

Criteria	AI Models	
	ML Models	DL Models	
Complexity	Moderate	High	
Explainability	Moderate	Low	
Potential in modern AI apps	Broad	Extensive	
Volume of data	Moderate to High	High	
Accuracy	Moderate to High	High	
Ease of use	Moderate	Moderate to Low	
Speed	Moderate	Moderate to High	
Adaptability	Moderate	High	
Self-learning	Low to Moderate	Low to Moderate	
Scalability	Moderate	High	
Types of algorithms	Linear Reg, Logistic Reg, Decision Trees, Random Forest, SVM, k-NN, Naïve Bayes, Gradient Boosting, Bayesian, Ensemble Methods	Multilayer Perceptron, CNN, RNN, Transformers, Siamese Networks, Capsule Networks, Attention Mechanisms, Neural Style Transfer, Deep Reinforcement Learning	

Health coaching, a medical intervention aimed at guiding patients in improving their health behaviors, has been shown to reduce expenses associated with managing chronic conditions (Wan et al., 2023). AI approaches also help improve patients’ ability to manage their diseases, reducing unnecessary visits. Integrating AI with skincare coaching creates a comprehensive model, using sensors, scanners, or cameras to gather data, AI models to generate insights, and visual analytics tools to present data in user-friendly formats (Zhou et al., 2017; Shinde et al., 2023).

AI-based tools for skin cancer diagnosis typically follow five stages: image capturing, preprocessing, segmentation, feature extraction, and classification. Various methodologies and techniques have been identified in the literature for each of these stages. The essential tools and algorithms used at each stage are summarized in Fig. 6, with Table 3 providing an overview of the methodologies used in skin cancer detection. In Fig. 6, the ‘feature extraction and selection’ block applies specifically to traditional ML methods, where features are manually extracted and selected by the user. However, when using deep learning networks, such as convolutional neural networks (CNNs), the feature extraction process is inherently performed by the network itself. These deep models learn hierarchical features directly from the input images during training, with no manual intervention required for feature extraction. To reflect this distinction, the framework applies ‘feature extraction and selection’ only in cases involving classical ML techniques.

Figure 6 Generic framework for AI-based skin cancer detection system.

Table 3 Summary of AI techniques used in different stages of skin cancer detection.

Ref	Year	Dataset and no of classes	Image preprocessing	Segmentation	Feature extraction	Classification	Optimizers	Performance metrics (%)	
Deep Learning	
Asiri et al. (2023)	2023	ISIC (seven classes)	Gaussian Filtering	Multilevel OTSU Thresholding, Swallow Swarm Optimization Algorithm	MobileNetV2	Deep Wavelet NN	Emperor Penguin Optimizer	Sensitivity	97.27	
Specificity	99.52	
Accuracy	99.19	
F1-score	97.24	
		
Shinde et al. (2023)	2023	ISIC (two classes)	Black-Hat filter	Threshold		Squeeze-MNet	Adam	Accuracy	99.36	
Sethanan et al. (2023)	2023	HAM10000 (seven classes)		Double-AMIS		Ensemble CNN		AUC,	99.4	
F1-Score	99.1	
MB (two classes)	Accuracy	98.8	
Venugopal et al. (2023)	2023	ISIC 2020, ISIC 2019, HAM 10000	Image blurring, distortion, resizing, normalization		Fine-tuned DNN	Modified EfficientNetV2-M	Adam	Accuracy of the different datasets	99.23	
97.06	
95.95	
Kumar et al. (2023)	2023	Kaggle Re-Snet50 datasets		ROI	AuDNN	DNN		Accuracy	93.26	
Mridha et al. (2023)	2023	HAM 10000	Normalisation			CNN+ XAI	Adam	Accuracy	82	
RMSProp	
Gururaj et al. (2023)	2023	HAM10000	Dull razor	Auto encoder-decoder		DenseNet169, Resnet 50		Accuracy	91.2 (Dense)	
83 (ResNet)	
Lee & You (2023)	2023	CelebA-HQ	Dull Razor	U-Net with dual encoding Backbones	Transfer backbone, CNN Backbone	GAN	Adam	PSNR		
ISIC 2020	SSIM	
FID	
LPIPS	
Elshahawy et al. (2023)	2023	HAM10000	Data Augmentation			YOLOv5+ResNet50	Top-notch	Precision	99	
Recall	98.6	
DSC	98.8	
Accuracy	99.5	
Ogundokun et al. (2023)	2023	ISIC	Data Augmentation			SkinNetX	SGD	Accuracy	97.56	
Behara, Bhero & Agee (2023)	2023	ISIC2017	Bicubic Interpolation, High-Pass filter, USM, Color transformation, Median Filter			DCGAN	SGDM	Accuracy	99.38	
Abbas et al. (2023)	2023	PSL	Color space transformation and contrast adjustment			SqueezeNet-Lightweight	SGD	Sensitivity	94	
Specificity	96	
Accuracy	95.6	
Precision	94.12	
F1-Score	95.2	
Wang et al. (2023)	2023	ISIC2018	Resize, Hair Removal, Data augmentation			DenseNet-121 + VGG16		Accuracy	91.24	
Malibari et al. (2022)	2022	ISIC2019	Wiener Filtering	U-Net	SqueezeNet	ODNN-CADSCC	IOWA	Accuracy	99.90	
Lucieri et al. (2022)	2022	ISIC2019				EXAID		Accuracy	100	
PH2, derm7pt	Precision	100	
Recall	100	
AUC	100	
Rasel, Obaidellah & Kareem (2022)	2022	PH2				CNN		Accuracy	97.50	
Precision	98	
Sensitivity	98	
Razzak & Naz (2020)	2022	ISIC2018				Deep Shallow Unit-Vise RNN		Accuracy	98.05	
Wang et al. (2022)	2022	HAM10000				Interpretability-Based Multimodal CNN		Sensitivity	72	
AUC_SEN_80	21	
Sharma et al. (2022)	2022	HAM10000			Handcrafted Features (Color moments +GLCM)	Cascaded Ensemble Deep Learning		Accuracy	98.3	
Alahmadi & Alghamdi (2022)	2022	ISIC2017		UNet	Feature fusion	CNN	SGD	Accuracy	95.91	
ISIC2018	95.91	
PH2	97.11	
Okuboyejo & Olugbara (2022)	2022	ISIC Archive				Ensemble Deep Learning (SMVE, WMVE)		Accuracy	95.74	
Ali et al. (2021)	2021	HAM10000	Gaussian Blurring, Median Blurring, normalization	Data Augmentation		DCNN		Accuracy AUC	93.16	
84.6	
Abdar et al. (2021)	2021					Bayesian Deep Learning	Bayesian Optimizer	Accuracy F1-Score	90.96	
91.00	
Sayed, Soliman & Hassanien (2021)	2021		Random over-sampling method	Data Augmentation		Squeeze Net	Bald Eagle Search Optimizer	Accuracy	98.37	
Specificity	93.47	
Sensitivity	100	
F1-Score	98.40	
AUC	99	
Thurnhofer-Hemsi et al. (2021)	2021	HAM10000				Shifted MobileNetV2 + GoogLeNet		Accuracy	83.2	
Bian et al. (2021)	2021	ISIC2017 (two classes)		Wavelet transformation	LBP	Multi-view Filtered Transfer Learning (MFTL)		AUC	91.8	
Machine Learning	
Priyadharshini et al. (2023)	2023	DermIS dataset (two classes)	Median Filter, Contrast enhancement	FCM	PCA	ELM-TLBO	TLBO	Accuracy	93.18	
Precision	89.72	
Recall	92.45	
F1-Score	91.64	
Ahammed, Mamun & Uddin (2022)	2022	ISIC 2019	Black Hat Transformation	Grab cut	GLCM	Decision Tree		SVM Accuracy	95 (ISIC), 97 (HAM)	
HAM10000	SVM	
KNN	
Alwakid et al. (2022)	2022	HAM10000	Data Augmentation			Modified XceptionNet		Accuracy	100	
Sensitivity	94.05	
PR	97.07	
F1-Score	95.53	
Cheong et al. (2021)	2021		Image texture enhancement	Bi-dimensional Empirical Mode Decomposition (BEMD)	Entropy and energy feature mining	SVM+Radial Basis Function		Accuracy	97.50	
Jiang, Li & Jin (2021)	2021	Data from northeastern China	RGB Color Space, Data Augmentation			DRANet+Squeeze-ExcitationAttention+CAM		Accuracy	86.8	
Machine Learning + Deep Learning	
Maqsood & Damaševičius (2023)	2023	HAM10000, ISIC2018, ISIC2019, PH2	BIMEF	26-layered CNN	Xception, ResNet-50, ResNet-101 VGG16	SVM	Poisson distribution	Accuracy of the different datasets	98.57, 98.62, 93.47, 98.98	
Suiçmez et al. (2023)	2023	HAM10000 (seven classes)	Colour Transformation, Black-Hat	Binary + Otsu + wavelet transform with visushrink	CNN	Hybrid CNN-GBC, Hybrid CNN-Machine Learning	Adam	Accuracy	99.4	
Precision	100	
ISIC 2020	Recall	
F1-Score	
Magdy et al. (2023)	2023	ISIC Archive	Median Filter	Segmented	GLCM	KNN-PDNN	Grey wolf Optimizer	Accuracy	99.375	
AlexGWO	99	
Ravi (2022)	2022	HAM10000				EfficientNetV2		Accuracy	99	
SVM	
RFTree	

In recent years, total body or whole body imaging systems integrated with AI have shown significant advancements in skin cancer prevention. These systems enable large-scale monitoring and early detection of lesions, which is crucial in preventing skin cancer (Korotkov et al., 2019; Birkenfeld et al., 2020). Studies have highlighted how AI-enhanced systems can track changes over time, improving melanoma and skin abnormality detection, thereby reducing missed diagnoses (Strzelecki et al., 2021; Soenksen et al., 2021).

Research on skin cancer detection also explores the integration of IoT technology with deep learning models for skin lesion diagnosis. IoT enhances healthcare environments by ensuring seamless connectivity and optimizing resource usage in healthcare systems, though it raises concerns regarding privacy and security during data transmission (Asiri et al., 2023). CNNs with adaptable activation functions are also used for skin lesion classification, offering flexibility for different lesion types. However, the complexity of selecting optimal activation functions increases training costs (Rasel, Obaidellah & Kareem, 2022; Albahar, 2019).

A unit-vice model has been applied for skin cancer classification, with a transition layer combining deep and shallow networks to capture diverse features. However, the increased depth of the model raises the risk of overfitting (Razzak & Naz, 2020).

Pham et al. (2020) propose a hybrid method to address class imbalance in skin disease classification using a dataset of 24,530 dermoscopic images. Their approach combines balanced mini-batch logic and real-time image augmentation with a new loss function. While the method shows promise, its reliance on a single large dataset and implementation complexity may limit generalizability in clinical settings. Further research is needed to validate its effectiveness across diverse datasets and real-world applications.

Kumar et al. (2022a, 2022b) propose using the Fractional Student Psychology Optimization (FSPBO)-based Deep Q Network (DQN) for skin cancer detection in a wireless network scenario. This method demonstrates high accuracy, sensitivity, and specificity but may face challenges due to the complexity of the multiple algorithms involved, which could limit its real-world applicability.

Duong et al. (2022) trained a neural network classifier on approximately 100 images per condition to classify six genetic conditions. The classifier outperformed paediatricians and geneticists in accuracy, suggesting that neural networks have strong potential in healthcare. However, further exploration in clinical settings is needed.

Yao et al. (2022) investigated the effectiveness of a single deep-learning model on imbalanced small datasets. Although the model addressed data limitations, it risks overfitting due to the small dataset size. Similarly, a hybrid approach combining CNN and transformer features (Alahmadi & Alghamdi, 2022) leveraged both labelled and unlabeled data effectively but increased model complexity.

Tang et al. (2020) addressed inter-class similarity and intra-class variation in skin lesion classification by proposing the Global-Part Convolutional Neural Network (GP-CNN). While the model achieved state-of-the-art performance on the ISIC 2016 and 2017 datasets, its complexity poses challenges for practical implementation.

Digital or real-time images from data centers or medical diagnostic labs are acquired in the first stage. Once captured, the images are preprocessed and segmented using various methods such as thresholding, region-based, feature-based, and edge-detecting approaches (Zhang et al., 2019). After segmentation, feature extraction and selection for the most relevant skin lesion features are performed, followed by classification using machine learning or deep learning classifiers (Behara, Bhero & Agee, 2023).

Different preprocessing techniques, segmentation processes, feature extraction, classification techniques, and optimizers are crucial for enhancing model performance in skin cancer detection. Techniques like Gaussian filtering (Asiri et al., 2023), black hat filtering (Shinde et al., 2023), Wiener filtering (Malibari et al., 2022), and median filtering (Shinde et al., 2023) reduce noise and artifacts. Additionally, image enhancement methods and data augmentation improve data quality (Behara, Bhero & Agee, 2024a). Preprocessing plays a vital role in optimizing model performance.

For segmentation, techniques like multilevel OTSU thresholding and region of interest (ROI) are commonly used (Kumar et al., 2023; Talavera-Martínez, Bibiloni & González-Hidalgo, 2021; Ashraf et al., 2020). Efficient feature extraction methods, including MobileNetV2 (Asiri et al., 2023; Akay et al., 2021), Xception, ResNet-50, ResNet-101, and VGG16 (Maqsood & Damaševičius, 2023), as well as features like local binary pattern (LBP) (Bian et al., 2021), principal component analysis (PCA) (Priyadharshini et al., 2023), and Gray Level Co-occurrence Matrix (GLCM) (Magdy et al., 2023), help reduce data dimensionality and improve model accuracy.

Pacheco & Krohling (2021) use deep learning to enhance skin cancer classification by combining images and patient metadata. They introduce the Metadata Processing Block (MetaBlock), which integrates metadata to emphasize relevant features from skin lesion images. This approach improves classification accuracy by considering visual and demographic data like human experts. MetaBlock outperformed MetaNet and feature concatenation methods across two datasets in 6 out of 10 scenarios. While promising, its effectiveness in diverse clinical settings needs further validation (Pacheco & Krohling, 2021).

Talavera-Martínez et al. (2022) developed a CNN model to classify skin lesion symmetry. While the model outperforms traditional methods, the small dataset of 615 images highlights the need for improvement. The study’s limited dataset may affect generalization to diverse populations. Additionally, the potential benefits of various data augmentation techniques (Putra, Rufaida & Leu, 2020) and alternative model architectures were not fully explored. Further research is needed to validate and enhance the model’s robustness and accuracy in broader clinical settings. Imran et al. (2022) address skin cancer detection using an ensemble of deep learning models (VGG, CapsNet, and ResNet) to improve prediction accuracy on the ISIC dataset. Traditional machine learning requires labour-intensive feature engineering, while deep learning offers automatic feature extraction. Their ensemble approach enhances accuracy, sensitivity, specificity, F-score, and precision compared to individual models (Goyal et al., 2020; Wei, Ding & Hu, 2020). However, the study’s reliance on the ISIC dataset raises concerns about generalizability (Zhao et al., 2021). The complexity and resource intensity of implementing an ensemble of deep learning may limit clinical applicability, necessitating further exploration of computational costs and validation in real-world settings (Imran et al., 2022). Tae et al. (2021) investigated applying low-shot deep-learning models for detecting conjunctival melanoma using a small dataset of ocular surface images. The dataset consisted of images from four classes: conjunctival melanoma, nevus or melanosis, pterygium, and normal conjunctiva. Two generative adversarial networks (GANs) were used to augment the training dataset. The dataset was divided into training, validation, and test sets, and 3D melanoma phantoms were created for external validation. Various deep learning architectures were employed, including GoogleNet, InceptionV3, NASNet, ResNet50, and MobileNetV2. MobileNetV2 achieved the highest accuracy with GAN-augmented images, demonstrating the potential of low-shot learning for accurate melanoma detection using smartphone images (Tae et al., 2021; Alwakid et al., 2022).

A hybrid architecture proposed by Biasi et al. (2022), combining Cloud, Fog, and Edge Computing, shows promise for improving melanoma detection services and reducing retraining times, but several challenges remain. The proposed system’s complexity may pose significant implementation and maintenance difficulties, particularly in resource-constrained environments. Additionally, the reliance on continuous retraining to achieve robust models may lead to high computational costs and energy consumption. The study also primarily focuses on the technical aspects, with limited discussion on integrating such systems into existing clinical workflows and ensuring user-friendliness for medical practitioners. Despite these limitations, the research underscores the importance of scalable and adaptable systems for managing large datasets and ensuring accurate and timely melanoma detection (Biasi et al., 2022).

Adegun & Viriri (2020) propose a framework for skin lesion detection and classification using deep learning, incorporating an encoder-decoder FCN and a DenseNet framework with CRF modules for enhanced feature learning and boundary localization. While the model shows high accuracy on the HAM10000 dataset, it faces challenges such as high computational demands, risk of overfitting, and limited generalization across diverse datasets. Additionally, the framework’s complexity and dependency on large, high-quality datasets could hinder its practical implementation in resource-limited settings. The study also lacks focus on model interpretability, which is crucial for clinical adoption (Adegun & Viriri, 2020).

Optimizers also fine-tune the classifiers that improve the model’s performance (Tan, Zhang & Lim, 2019; Sayed, Soliman & Hassanien, 2021). Classifiers like Squeeze-MNet, Ensemble CNN, DNN, GAN, and SkinNetX are some classification methods used for efficient detection, as presented in Table 4. The ensemble method improves performance more than single classifiers. Their strengths help show improved performance even though most methods show accurate detection results with improved performance. These methods also suffer from various limitations, like the need for more computational resources, increased complexity due to more models, overfitting due to small-size datasets, and inaccurate detection with more errors. This review helps to understand the insights of various existing methods in skin cancer detection.

Table 4 Comparison of traditional dermatological methods vs. AI methods.

Metric	Traditional Methods	AI Methods	
Diagnostic accuracy	75–80% for dermoscopy (Dinnes et al., 2018)	90–95% with AI (Esteva et al., 2017; Haenssle et al., 2018)	
Speed	Days to weeks for biopsy and histopathology (Kaplan & Haenlein, 2020)	Real-time diagnostic feedback (Esteva et al., 2017; Xie et al., 2020)	
Accessibility	Requires specialized dermatologists and equipment (Argenziano et al., 2021)	AI tools can be accessed remotely via telemedicine (Xie et al., 2020; Brinker et al., 2019b)	
Scalability	Limited by clinician availability (Swetter et al., 2019)	Scalable with continuous improvement through new data (Tschandl et al., 2020; Gandomi, Chen & Abualigah, 2022)	
Bias	Prone to human bias and variability (Brinker et al., 2019a)	Less variability and more objective assessments (Buolamwini & Gebru, 2018)	

The review highlights significant differences between classical machine learning systems, where features are manually selected using techniques like the Gray Level Co-occurrence Matrix (GLCM), and deep learning systems, where neural networks automate feature extraction. Classical systems, such as those using GLCM, often rely on handcrafted features that may fail to capture the complexity of skin lesion textures, shapes, and color variations. While these methods have performed well on smaller datasets, their limitations become apparent when applied to more complex lesions or larger datasets (Soenksen et al., 2021).

In contrast, deep learning-based systems, especially CNNs, excel in automatically learning and extracting hierarchical features from raw images, often outperforming classical methods in terms of diagnostic accuracy, particularly in multiclass classification tasks involving diverse lesion types (Anjum et al., 2020). A key strength of deep learning systems is their ability to adapt to new data without the need for manual feature engineering, which is time-consuming and subject to bias. However, deep learning systems’ increased computational complexity and resource requirements pose challenges for their implementation in resource-constrained settings (Ahmad et al., 2021).

Using existing literature, we comprehensively analysed ML and deep learning (DL) models to assess their ability to generalize and perform well in identifying skin cancer. This analysis employed ten criteria, as specified in Table 4. The criteria include (1) Complexity, (2) Expandability, (3) Potential in modern AI applications, (4) volume of data required for the AI models, (5) accuracy, (6) ease of use, (7) speed, (8) adaptability, (9) self-learning, and (10) scalability. Based on the defined ten criteria, DL models remain highly adaptable and complex to the medical AI application, as shown in Fig. 7.

Figure 7 Classification of AI models according to their level of complexity and adaptability.

Dimension 4: Functionalities of AI systems

This study explores how artificial intelligence (AI) provides medical organizations with a range of functionalities, including clinical decision-making, resource allocation, and information sharing. AI also supports patients through diagnosis, treatment, consultation, and health monitoring. Additionally, AI offers industry-level functionalities such as IoT data gathering, medical imaging, research development, and remote surgery (Sethanan et al., 2023; Maqsood & Damaševičius, 2023). The subsequent sections will provide a detailed examination of each category.

Functionality features for the industry

1. Data acquisition

Utilizing a multidisciplinary approach that incorporates IoT (Gandomi, Chen & Abualigah, 2022; Tobore et al., 2019) with skin lesion image processing and research development, AI can be effectively leveraged across dermatological departments and sectors (Behara, Bhero & Agee, 2023). Studies have highlighted how AI and IoT drive advancements in the medical sector by transmitting patient pharmacological information and history, along with environmental data, to cloud platforms for enhanced medical analysis (Morales, Engan & Naranjo, 2021).

2. Accurate diagnosis

Accurate diagnosis is essential for implementing effective treatment, particularly in the early stages of illnesses (Ali et al., 2021). Early detection can save significant time, effort, and financial resources for patients, dermatologists (Brinker et al., 2019a), and skincare organizations. For example, the 5-year survival rate for melanomas under 1 mm in thickness is 95%, while for those exceeding 4 mm, it drops to 45% (Behara, Bhero & Agee, 2024b). AI systems must be validated against clinical standards by comparing AI-generated diagnoses with those of experienced dermatologists and histopathological findings. This ensures AI systems meet necessary accuracy and reliability benchmarks for clinical application (Brinker et al., 2019b).

3. Usage of AI

The use of AI to quantify and identify physical abnormalities significantly impacts mortality rates, particularly through early diagnosis (Frederico & Krohling, 2022). Early detection of cancer, especially before metastasis, dramatically improves the chances of therapeutic success and survival. Researchers have developed models, such as the AuDNN, which predicts skin cancer risk by combining backpropagation gradient descent with least squares for feature reduction, aiming to shorten diagnostic time. This model, when tested with fluorescence images, demonstrated strong accuracy compared to existing systems (Kumar et al., 2023). Many studies have utilized AI for early skin cancer detection, showing promising results (Rahman et al., 2020).

Researchers have conducted comparative analyses using various classifiers, with CNN systems showing strong performance in identifying skin conditions from skin disease datasets. This technology aids in precise diagnosis and treatment recommendations (Mridha et al., 2023; Song et al., 2020).

Hybrid AI-human decision-making approaches are gaining prominence in medical diagnostics, including skin cancer detection. These systems enhance AI’s computational power with the clinical expertise of healthcare professionals, allowing for more reliable diagnoses. AI can flag high-risk lesions while dermatologists make the final decision, improving the overall diagnostic process (Esteva et al., 2019). Explainable AI (XAI) methods provide transparency, allowing human experts to understand and validate AI predictions, which fosters trust in clinical settings (Gunning et al., 2019). Studies have demonstrated that AI-human collaboration often outperforms both AI-only and human-only diagnoses, particularly in complex cases requiring nuanced interpretation (Tschandl et al., 2020). This hybrid model bridges current AI limitations, making it more adaptable to real-world clinical applications and improving patient outcomes (Mridha et al., 2023).

Education for patients and doctors

AI algorithms in dermatoscopy hold great promise, but their integration into clinical practice presents several challenges that extend beyond technical limitations. These challenges encompass education for clinicians and patients, legal considerations, and regulatory issues that must be addressed for safe, reliable use in clinical diagnosis. One of the critical limitations of AI in dermatoscopy is the lack of education for both patients and clinicians regarding the proper use of these systems. Patients may perceive AI as infallible, leading to unrealistic expectations of accuracy. For clinicians, specialized training is essential to understand how AI works and how its outputs should complement their judgment. Studies have shown that AI enhances diagnostic accuracy when used correctly, but over-reliance without proper understanding could result in misdiagnoses (Esteva et al., 2017). Education programs that highlight AI’s role as an assistive tool rather than a replacement are crucial for safe integration into clinical workflows (Han et al., 2020).

Legal and ethical considerations

Legal issues surrounding the use of AI in clinical settings are multifaceted. Accountability remains a crucial concern: if an AI misdiagnoses a patient, it is unclear whether liability falls on the developer, the healthcare provider, or the institution (Rieke et al., 2020). Additionally, data privacy is a significant concern, particularly under regulations like the General Data Protection Regulation (GDPR) in the EU and HIPAA in the US, which impose strict guidelines for handling sensitive medical data (European Commission, 2020). AI systems must comply with these regulations, ensuring that patient data is anonymized and secure. Furthermore, bias in AI models trained on non-representative datasets can result in unequal care, particularly for underrepresented groups, raising severe ethical concerns (Buolamwini & Gebru, 2018).

Regulatory challenges and clinical validation

The regulation of AI systems in dermatoscopy is critical for ensuring reliability and safety in clinical practice. Many AI systems rely on proprietary datasets that may not undergo the rigorous scrutiny necessary for clinical deployment. Regulatory bodies such as the FDA in the US and the EMA in Europe are working to develop standards for approving AI in healthcare. However, comprehensive global guidelines are still lacking (FDA, 2021). Moreover, AI systems require continuous validation and monitoring to ensure their efficacy over time, especially as they learn from new data. Adaptive algorithms must undergo regular re-validation to ensure they remain safe and effective in diverse patient populations (Mandl & Kohane, 2012).

Clinic-specific functions

AI technologies can assist in analyzing medical imaging and early disease identification while also preventing human error-related misdiagnoses (Venugopal et al., 2023). AI has the potential to be applied across multiple domains in the medical industry, benefiting clinicians, patients, and other stakeholders. Clinicians can leverage technological advancements, contemporary information collection methods (Frederico & Krohling, 2022; Lucieri et al., 2022; Combalia et al., 2022), and effective information-sharing practices to make more informed decisions. Machine learning, in particular, enhances decision-making by uncovering latent information from vast datasets that would be difficult for humans to analyze manually. Studies have shown how machine learning augments and automates decision-making in the skin cancer sector (Cheong et al., 2021). By applying AI to skin cancer classification, the reliability and accuracy of AI-assisted diagnoses can improve, providing essential data to support medical practitioners in their treatment choices (Mridha et al., 2023).

Patient functionality

Health interventions primarily focus on enhancing patient functionality, and AI technology offers a wide range of applications, from efficient appointment scheduling to patient monitoring (Patel et al., 2021; Jiang, Li & Jin, 2021). In dermatology, AI plays a significant role in patient diagnostics (Carvalho et al., 2021), consultation, treatment (Han et al., 2020), and health monitoring (Hossain, Ferdousi & Alhamid, 2020). Hekler et al. (2019) developed a wearable telehealth device to facilitate patient monitoring. By reducing false alerts, AI helps decrease hospital loads, resource consumption, and unnecessary interventions, allowing healthcare professionals to focus on more critical tasks. Remote patient monitoring offers advantages for older patients, minimizing unnecessary hospital visits and enhancing patient safety and well-being through timely medical alerts (Sondermann et al., 2019). Experts recommend AI integration in medical processes to address the increasing patient population and optimize biomedical data management (Burlina et al., 2019).

Summary of literature

Our systematic literature review of AI in skin cancer using the four-dimension classification framework has yielded the following insight: AI systems can detect diseases early with enhanced diagnostic precision, facilitating faster diagnosis and therapy decisions by assisting dermatologists in clinical decision-making.

AI tools can be broadly and readily deployed to provide remote skin lesion screening and assessment for potential treatment via image submission and analysis.

The integration of AI model tools into a clinical setting and the availability of high-quality data for training, interpretability, data privacy and security, and cost are all challenges that AI systems must overcome.

AI models primarily focus on image-based skin cancer diagnostic techniques, including machine learning, deep learning, and computer vision models.

Integrating AI technology within the medical field has significantly aided dermatologists in their diagnostic workflows, facilitated remote dermatological consultations, and enhanced patient education regarding treatment options.

Study inference

The empirical studies that account for the rapid commercial integration of AI solutions in early skin cancer detection are currently at an early stage of development. Cutting-edge AI technologies are currently employed as the next generation of health informatics and digital health solutions. Multiple information technology (IT) companies now provide these advanced AI technologies. While integrating AI components is a recent development, utilizing digital technologies in organizational and medical processes within skin cancer detection is not novel.

Theoretical implications

Theoretical studies on AI applications in the skincare business should adopt a cross-theory approach, analyzing data from individual (patient and medical staff), organizational (hospital and team), and sectoral (dermatology industry) perspectives (Jiang, Li & Jin, 2021). AI offers benefits at every level, from personal advantages for patients and medical personnel to organizational efficiencies for hospital administrators.

In terms of accuracy, efficiency, and speed of medical and administrative activities, AI consistently outperforms humans, significantly impacting the future direction of skin cancer diagnostics (Das et al., 2021). For patients, AI enhances clinical safety, patient experience, and holistic care. For medical professionals, AI accelerates research on medical treatments, improves diagnostic accuracy, and enables advancements like robot-assisted surgery (Carvalho et al., 2021). AI systems are reshaping skincare by learning and adapting to complex diagnostic challenges.

However, AI in dermoscopy presents issues related to data integrity, patient safety, and privacy. Patient scenarios often involve variables that are difficult to capture digitally, which can limit the real-world applicability of AI-driven diagnoses. Moreover, “black-box” AI systems are challenging to interpret, leading to concerns about the lack of transparency in how diagnoses are reached. Clinicians must understand how and why AI models generate specific outcomes to ensure trust in these systems. Without accountability, AI solutions may face resistance from medical professionals (Hossain, Ferdousi & Alhamid, 2020).

AI’s role in healthcare continues to evolve, supported by advances in machine learning algorithms and transparent data integration methods that protect patient privacy. Its benefits span across diagnosis, treatment, consulting, and chronic disease management (Swetter et al., 2019). AI supports clinical decision-making in medical imaging and robotic surgery, mainly through autonomous IoT data collection, which enhances the reliability of skin cancer diagnoses by capturing critical data points (Mobiny, Singh & Van Nguyen, 2019).

Our findings align with previous reviews demonstrating the positive impact of AI-driven tools on dermatological diagnostic accuracy and workflow efficiency (Deepa, ALMahadin & Sivasamy, 2024; Gohil & Desai, 2024). However, challenges such as data scarcity and slow clinical adoption remain, highlighting the need for interdisciplinary collaboration and the development of robust datasets for AI integration (Strzelecki et al., 2024; Furriel et al., 2024).

Practical implications

AI reduces clinical trial costs by streamlining processes and saving human hours in researching new treatments for skin cancer. It optimizes patient interaction workflows, addressing complexities related to co-morbidities, insurance, and environmental factors, thus enhancing the patient experience. AI effectively integrates critical skincare data for diagnosis, treatment, and prevention at an organizational level (Ravi, 2022). However, AI applications in skincare face challenges like insufficient diagnostic error testing, which raises concerns about data integrity and ethical issues in medical data collection. Configuring an AI training model involves processing thousands of images, which can be time-consuming and prone to errors.

Additionally, skin lesion data is sensitive and complex, making it vulnerable to breaches and cyberattacks. AI systems used in skincare require high-quality data processing and automation. While medical images and videos need expert interpretation, textual data (such as notes and reports) is easier to handle. Nevertheless, processing complex multimedia can slow down AI systems and compromise diagnostic accuracy, posing risks to patient safety.

Conclusion and future scope

AI has great potential to revolutionize the field of skin cancer diagnosis. The potential benefits of this innovation, such as its accuracy, ease of use, efficiency, and cost-effectiveness, have the potential to improve early detection of skin cancer and save many lives. However, it is crucial to carefully address and ethically implement difficulties like data bias, false positives, explainability, and limitations to ensure that AI’s function is equally and effectively beneficial to everybody. The ongoing research and development in AI indicates a promising future for its application in skin cancer detection. This technological advancement will enable a more alert and proactive approach towards combating this widespread disease. In Dermatology, machine learning, IoT, algorithms, and robotics are used to monitor, diagnose, treat, and quantify risks and benefits. Skin lesion data help skincare organizations improve operations and administer medicine. This systematic review examines the advantages, challenges, techniques, and capabilities of AI in detecting skin cancer. This review found that AI and its subfields assist individuals, organizations, and dermatologists. Data integration, Privacy, legal issues, and patient safety are difficulties.

This systematic study found that AI uses machine learning, image processing, data mining, expert systems, virtual reality, computer vision, and data science. AI can diagnose, treat, share, secure, consult, monitor, collect data, and perform remote surgery. The review highlights its potential to revolutionize dermatology, offering improved accuracy, efficiency, and cost-effectiveness, ultimately leading to early detection and potentially saving lives. Due to high skincare standards, AI for skin cancer detection must be thoroughly reviewed for data integrity, patient safety, and privacy issues. Thus, many organizations are updating patient data security rules to ensure confidentiality. Stricter regulations and updated security protocols are necessary to safeguard patient information and prevent unauthorized access or breaches. AI should be regarded as a beneficial tool that supports and enhances human expertise rather than supplanting it. The primary objective is to establish a robust collaboration between artificial intelligence and dermatological professionals to provide optimal patient treatment.

Future work will focus on enhancing data integrity, privacy measures, and collaboration between AI and dermatological professionals to optimize patient treatment and outcomes. Additionally, efforts to update patient data security rules and establish robust collaborations between AI and dermatology professionals will be essential for successfully integrating AI into skin cancer detection and treatment.

Supplemental Information

Supplemental Information 1 Categorization of literature based on classification framework.

The data presented in this study are openly available in the reference list. I sincerely thank my supervisor, Ernest Bhero, and co-supervisor, John Terhile Agee, for their excellent supervision and assistance in paying attention to detail. Moreover, I wish to thank my family for their continuous support and countless sacrifices.

Additional Information and Declarations

Competing Interests

Author Contributions

Data Availability

The authors declare that they have no competing interests.

Kavita Behara conceived and designed the experiments, performed the experiments, analyzed the data, performed the computation work, prepared figures and/or tables, authored or reviewed drafts of the article, and approved the final draft.

Ernest Bhero conceived and designed the experiments, performed the experiments, analyzed the data, authored or reviewed drafts of the article, and approved the final draft.

John Terhile Agee performed the experiments, analyzed the data, authored or reviewed drafts of the article, and approved the final draft.

The following information was supplied regarding data availability:

This is a literature review.

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
