# Peer review of "AI in dermatology: a comprehensive review into skin cancer detection"

_PeerJ Computer Science, doi:10.7717/peerj-cs.2530_

## Round 0.1 · original submission · Major Revisions

Dear Author
Greetings for the Day

Good work, however, some corrections are required to make the article more effective . Please make the changes and re-submit it for review again

Thanks
Editor - PeerJ Computer Science

Reviewer 1 ·

Basic reporting

The reviewed paper provides an overview of automated methods for skin cancer detection in dermatological images using various artificial intelligence (AI) techniques. Given the rapid development of AI methods in dermatology, such reviews are highly necessary and can be a valuable resource for dermatologists interested in diagnostic support systems for skin diseases. However, several issues require clarification and supplementation before the article is suitable for publication.

1. The review's failure to include other review articles is a significant omission. These articles are crucial for analyzing the conclusions presented therein and comparing them to the conclusions drawn in the current work. See e.g.:

10.1063/5.0188187
10.1007/s11831-024-10121-7
10.1016/j.clindermatol.2023.12.022
10.3389/fmed.2023.1305954

2. The review should also consider studies that describe Total Body or Whole Body Systems, the importance of which has recently been increasing, especially in the case of skin cancer prevention, see e.g.:

10.1109/JBHI.2018.2855409
10.1016/j.cmpb.2020.105631
10.3390/s21196639
10.1126/scitranslmed.abb3652

3. The discussion lacks conclusions drawn from the review of various skin lesion analysis systems.

• The review should acknowledge that these systems often detect multiple types of cancer, and their effectiveness varies depending on the type of skin lesion.
• For studies using proprietary datasets, the discussion should address the size of these datasets and whether they are sufficiently large to ensure the reliability of the results.
• The methods used to verify AI algorithms in different studies should be discussed to assess the reliability and generalizability of the results.
• The effectiveness of classical systems (where features are selected using algorithms like GLCM) and deep learning systems (where features are extracted by the network) should be compared.

4. The review includes an excessive number of figures related to the selected journals. One or two figures would be sufficient.

5. The Fig. 6 illustrating the general scheme of skin lesion image analysis is overly simplified and inaccurate. Image restoration is part of image enhancement, with hair removal algorithms being a particular example belonging to the latter group. Lesion segmentation can be done by deep networks before analyzing the mole. Feature selection is only needed for classical algorithms. If deep networks are used, the features are determined internally by the networks. This figure must be corrected.

6. The discussion of the limitations of AI algorithms in dermatoscopy is too general and superficial. The role of education for both patients and doctors should be emphasized. The legal aspects of using these algorithms in clinical diagnosis should be discussed in more detail, including regulation of introducing of such algorithms in clinical practice.

7. The table in Supplement 2 containing a quantitative comparison of the performance of different skin lesion image analysis systems is central to this review. Therefore, it should be inserted and described in the main text of the work.

Experimental design

see comments above

Validity of the findings

see comments above

Additional comments
* * *
Cite this review as

Reviewer 2 ·

Basic reporting

Here are my comments and suggestions for improving the quality of the manuscript titled AI in Dermatology: A Comprehensive Review into Skin Cancer Detection:

1.The manuscript is generally well-written but could benefit from a thorough language check for clarity and fluency. For example, certain sentences are lengthy and could be simplified for better readability (e.g., lines 70-76).
2.The introduction does a good job of providing background but could benefit from a clearer articulation of the gap in the existing literature that this review aims to address. Including a specific objective for the review would help to focus the manuscript.
3.Ensure that all figures are properly labeled and referenced in the text. It is important to verify that the images are not overly manipulated. Some of the figures (e.g., Figure 1) could be improved for better visual clarity.

Experimental design

4.While the methodology section uses the PRISMA guidelines, the search strategy and selection process should be explained in more detail. For example, specify whether studies from other relevant databases, like PubMed, were considered and provide more transparency on inclusion/exclusion criteria beyond the basic outline.
5.Ensure that the methodology comprehensively covers all key dimensions of AI in skin cancer detection. There is a potential gap in discussing studies that focus on hybrid AI-human decision-making approaches.

Validity of the findings

6.Although the review provides a broad overview of AI applications in skin cancer detection, the novelty of this review is not fully clear. Highlight any unique angles, such as newer trends or a specific focus on underexplored AI techniques, that differentiate it from previous reviews.
7.While the authors acknowledge the challenge of data privacy and data availability, a more detailed analysis of how this specifically affects the reliability of the AI models discussed could enhance the paper's depth.
8.The conclusion should offer a more structured summary of unresolved challenges and future research directions.

Additional comments

9.The paper would benefit from a clearer comparison of AI methods versus traditional diagnostic methods. Discussing how AI specifically improves over these approaches with quantitative examples (where applicable) would strengthen the argument.
10.When discussing the performance of AI models, it would be useful to include standardized performance metrics such as accuracy, sensitivity, specificity, and F1 scores in a more systematic way. Consider adding a summary table that highlights key performance metrics of the models discussed.
11.Add more figures that visually explain AI workflows, algorithms, or model architectures used for skin cancer detection. This would help the audience better understand complex AI techniques.
12.Provide a summary table of the literature review. I recommend citing additional recent and relevant research to strengthen the manuscript.
13.In more detail, address the potential bias in training datasets (e.g., underrepresentation of certain skin types or populations), along with solutions to mitigate these biases.

Cite this review as

·

Basic reporting

Introduction section needs sufficient literature references (latest) to substantiate the following lines given by the author

154 AI systems offer personalized treatment recommendations based on a patient's medical history and
155 condition, improving overall health outcomes and patient satisfaction. AI technologies also
156 monitor patients' progress and predict potential complications, allowing for early interventions that
157 prevent health issues from escalating and leading to better long-term health management.

Experimental design

Well structured and semantically categorized survey article

Validity of the findings

The description of datasets used by the methodologies will provide more clarity about the content to validate the findings

Additional comments

Good survey article with many findings provided in well organized manner.
Minor Suggestions for better improvements like dataset description in validating the methodology findings and latest reference to substantiate the saying "AI systems offer personalized treatment recommendations" can be considered

Cite this review as

---

## Round 0.2 · Minor Revisions

Dear Authors,

Good Work!

Please find a few suggestions to make your article more effective .So please complete and re-submit for the final review.

Thanks

Editor-PeerJ Computer Science

Reviewer 1 ·

Basic reporting

Thank you for correctly taking into account all my raised in the review and appropriately supplementing the text. One issue requires improvement. In Fig. 6, presenting the framework for AI-based skin cancer detection system, there is a block "feature extraction and selection". This block appears only when we use classic ML methods. In the case of using deep networks, image features are determined internally by the network, without the participation and influence of the user on this process. Please comment on this appropriately in the text when describing this figure.

Experimental design

This is correct.

Validity of the findings

This is correct.

Cite this review as

Reviewer 2 ·

Basic reporting

Authors have addressed the comments and suggestions

Experimental design

Authors have addressed the comments and suggestions

Validity of the findings

Authors have addressed the comments and suggestions

Additional comments

Please merge the last two columns of Table 3 into one column

Cite this review as

·

Basic reporting

no comment

Experimental design

no comment

Validity of the findings

no comment

Additional comments

no comment

Cite this review as

---

## Round 0.3 · accepted · Accept

Dear Author/Authors

Good work! I have reviewed the final document and it is quite good. Please process for the next step.

Thanks
AE-PeerJ Computer Science